# Data Gatherer v0.2: High-Recall Retrieve-then-Read via Semantic Retrieval, Regex Anchoring, and Fine-Tuned T5

Pietro Marini
Data Tecnica International / New York University
pietro@datatecnica.com

Mike A. Nalls
Data Tecnica International / NIH Center for Alzheimer's Research

Mette Peters
Data Tecnica International / NIH Center for Alzheimer's Research

Nicole Contaxis
New York University

Juliana Freire
New York University

Aécio Santos
New York University

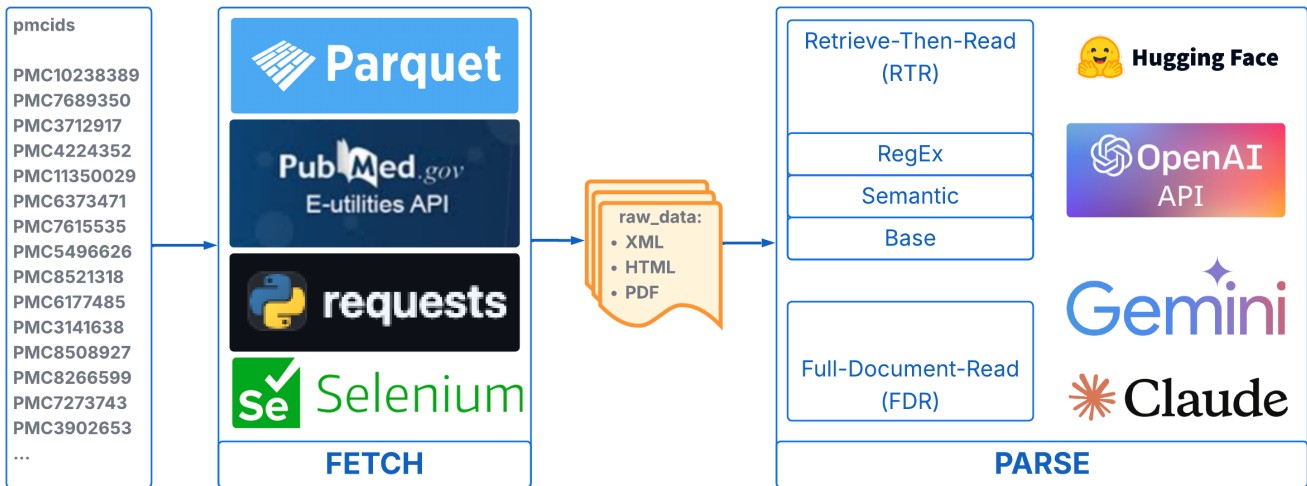

**Figure 1: Data Gatherer pipeline**

## ABSTRACT

We study automated extraction of dataset references from biomedical articles and address a key limitation of prior Retrieve-then-Read (RTR) pipelines: low recall when section retrieval fails to capture diverse citation practices. Building on our previous work [1], we introduce an enhanced RTR technique that combines semantic section ranking with ontology-guided regex anchoring, and we fine-tune a Flan-T5 model for zero-marginal-cost extraction at scale. On a held-out test split of 249 articles from the DataRef-REV benchmark, our full RTR pipeline raises Gold-GT recall from 0.48 to 0.92 and F1 from 0.35 to 0.58, spanning repositories beyond GEO and ProteomeXchange. We release model weights, training data, ontology, and evaluation code.

**PVLDB Reference Format:**
Pietro Marini, Mike A. Nalls, Mette Peters, Nicole Contaxis, Juliana Freire, and Aécio Santos. Data Gatherer v0.2: High-Recall Retrieve-then-Read via Semantic Retrieval, Regex Anchoring, and Fine-Tuned T5. PVLDB, 19(1): XXX-XXX, 2026.
doi:XX.XX/XXX.XX  **PVLDB Artifact Availability:**

Proceedings of the VLDB Endowment, Vol. 19, No. 1 ISSN 2150-8097.
doi:XX.XX/XXX.XX

The source code, data, and/or other artifacts have been made available at https://github.com/VIDA-NYU/data-gatherer.

## 1 INTRODUCTION

Linking papers to the datasets they cite is essential for reproducibility, data discovery, and large-scale analyses of scientific evidence. Yet dataset references in papers remain inconsistent and difficult to resolve: citations appear as accession codes, partial identifiers, informal names, or URLs dispersed across heterogeneous sections (e.g., data availability statements, supplementary material, methods, and figure captions) [1]. As a result, both manual curation and automated indexing remain incomplete.

In our prior work [1], we introduced Data Gatherer and evaluated two strategies for dataset reference extraction: Full-Document Read (FDR), which is robust but expensive, and Retrieve-then-Read (RTR), which reduces cost by selecting a subset of document sections before extraction. While RTR can match FDR on carefully curated articles, it degrades when retrieval rules fail to cover the diversity of real-world citation practices, leaving recall low on broader

benchmarks [1]. This recall–cost tension is the main obstacle to scaling paper–dataset linking across large corpora.

This paper strengthens RTR to achieve high recall without resorting to full-document LLM processing, and complements the system advances with a fine-tuned Flan-T5 model that enables large-scale runs at zero marginal API cost. Our contributions are grouped as follows:

(1) **System: high-recall RTR.** We introduce semantic section ranking and ontology-guided regex anchoring to robustly surface candidate evidence for extraction.

(2) **Model/data: open, low-cost extraction.** We release a training dataset and fine-tune Flan-T5 for dataset mention extraction, together with an anchoring audit that separates grounded outputs from entailed inferences and unsupported generations.

(3) **Evaluation: broader coverage signals.** We evaluate on a held-out 249-article split of DataRef-REV against two complementary reference signals: SciLite [4] annotations and a reverse engineered Gold standard.

(4) **Public release.** Model weights, training data, ontology, and evaluation code are released open-source.

## 2 SYSTEM

Figure 1 illustrates the two processing paths. Given a list of PMC article URLs, the system fetches, parses, and extracts dataset citations through a five-component pipeline. Both the **FDR** (Full-Document-Read) and **RTR** (Retrieve-then-Read) paths share the Fetcher and Parser stages; they diverge at the retrieval step. The FDR path passes the full parsed document to a large-context commercial API and is unchanged from our prior work [1]. The RTR path—the focus of this paper—selects a candidate section pool before calling the LLM, as described below. Both paths output a list of (`dataset_identifier`, `data_repository`) pairs per article.

### 2.1 Fetcher

The Fetcher accepts a list of PMC article URLs and retrieves the raw document content using up to three strategies in priority order. First, it submits a bulk request to the NCBI Entrez API, which returns structured XML for most PMC articles. If the XML output does not meet quality thresholds (minimum number of sections or presence of required section types), two fallbacks are attempted: (1) a lightweight HTTP GET request, and (2) a headless Selenium WebDriver for full-page rendering. The HTTP GET fallback is new relative to our prior work [1], which only used Selenium.

### 2.2 Parser

The Parser converts raw XML or HTML into an in-memory element tree. XML documents are processed with `lxml` under either the JATS or TEI schema; HTML documents are processed with BeautifulSoup.

### 2.3 Retrievers

The Retrievers build the candidate section pool that is fed to the LLM. Three sources contribute independently; their union is taken.

*Predefined patterns.* We include candidate sections using a rule-based retriever configured in an external patterns file.[1] The retriever applies a configurable allow-list of publisher- and layout-specific patterns (e.g., selectors/expressions targeting "Data Availability" and related blocks) and adds all matched sections verbatim to the candidate pool, as in our prior work [1].

*Semantic retrieval (new).* Each section is split into chunks that fit within the token window of a sentence-transformer model [3] (default: `all-MiniLM-L6-v2`) and embedded. Chunks are ranked by cosine similarity to a fixed query and the top-$k$ (default $k$=3) are added to the candidate pool.[2] Chunk embeddings are cached on disk to avoid recomputation across runs.

*Regex anchoring (new).* The full document text is scanned with regular expressions compiled from the `id_pattern` fields of the repository ontology (§2.5). Every section containing at least one match is added to the candidate pool, ensuring that accession IDs in unexpected section types are not missed.

The union of all three candidate sets is then passed as context to the LLM extraction step.

### 2.4 LLM Client

The LLM Client runs extraction over the candidate section pool using a model-agnostic interface. For the fine-tuned Flan-T5 [5] variant, a plain-text prompt is rendered and batched inference is run on a single GPU (`NVIDIA-H100-NVL`) at zero marginal API cost, making large-scale sweeps over PubMed Central practical. For commercial APIs (Claude, GPT, Gemini), provider-specific few-shot prompts request structured JSON output that additionally includes citation type (primary vs. secondary). Responses are normalised to (`dataset_identifier`, `data_repository`) pairs and deduplicated before output.

### 2.5 Data Repository Ontology

The repository ontology [3] is a curated JSON map of biomedical data repositories, keyed by either a repository domain (e.g., `massive.ucsd.edu`, `www.ebi.ac.uk`) or a short alias (e.g., geo, dbgap, pdb). Each entry provides operational metadata used by the pipeline, including a human-readable `repo_name` and a templated `dataset_webpage_url_ptr` with an `__ID__` placeholder. Many entries additionally define optional fields such as `id_pattern` (for accession matching when available), `download_root`, `access_mode`, `javascript_load_required`, and cross-entry mappings (e.g., `repo_mapping`, `repo_root`) to support repository-specific resolution and access. The ontology ships with the system and is updated iteratively as new repositories and identifier patterns are encountered.

## 3 FINE-TUNING FLAN-T5 FOR LOW-COST EXTRACTION

We fine-tune Flan-T5 to extract structured dataset references from short publication snippets, enabling batch inference on a single GPU and avoiding per-token API costs. Each training example consists of

---

[1] `data_gatherer/config/retrieval_patterns.json`

[2] Default query string defined in `data_gatherer/parser/base_parser.py`, function `semantic_retrieve_from_corpus`.

[3] `data_gatherer/config/open_bio_data_repos.json`

an input snippet and a target JSON array of (`dataset_identifier`, `data_repository`) pairs.

*Training data.* We release a large training dataset of PMC snippets paired with dataset mentions, constructed from DataRef-REV supervision and normalized to a common schema. The dataset is publicly available on Hugging Face.[4]

*Fine-tuning setup.* We train for a small number of epochs with gradient accumulation to fit on a single GPU, and standard seq2seq decoding with post-hoc normalization and deduplication.[5]

## 3.1 Anchoring Review (Examples)

During fine-tuning, we observed that some model outputs did not regex-match any accession pattern in the source snippet. To understand what these cases look like in practice, we manually reviewed a sample of flagged outputs and report representative examples in Table 1. The examples show that non-matches arise from (i) identifiers that are implicitly specified via ranges or lists, (ii) formatting variants in the source text (e.g., whitespace/hyphenation), and (iii) genuinely unsupported outputs such as placeholders or non-dataset identifiers.

**Table 1: Representative flagged examples (predicted identifiers not matching any accession regex in the source snippet).**

| Predicted ID | Snippet evidence (abridged) |
| --- | --- |
| GSM2816665 | Data Availability … "… series GSE105030 … samples GSM2816664 to GSM2816669." |
| GSE41306 | Data availability … "… (GSE 41306) … (GSE 217222) … deposited …" |
| E-MTAB-8556 | Data availability … "… accession number E-MTAB- 8556." |
| GSExxxxxx | Data and code availability … "… deposited at ncbi.nlm.nih.gov/geo/ …" |
| Yin et al. dataset | Materials and methods … "… data sets were downloaded … the Yin et al …" |
| No. 2021YFA1301601 | Funding … "… (2021YFA1302604 and 2021YFA1301601) …" |
| MsigDB v7.0 | Methods … "… Molecular Signatures Database (MsigDB) v7.0 …" |

## 4 EXPERIMENTAL SETUP

*Dataset.* We use the DataRef-REV benchmark [2] (1,242 PMC articles), reproduced with the same train/test split as the fine-tuning job (80/20, `random_state=42`): 993 training articles and a held-out test set of **249 articles** spanning GEO, PXD, dbGaP, Zenodo, Dryad, ArrayExpress, and dozens of other repositories.

*Retrieval configurations.* **base**—no semantic retrieval, no regex anchoring (replicates the prior system); **S3**—semantic retrieval ($k$=3), no regex anchoring; **RS3**—regex anchoring and semantic retrieval ($k$=3).

[4]https://huggingface.co/datasets/vida-nyu/pmc-articles-dataset-mentions-snippets
[5]https://huggingface.co/vida-nyu/flan-t5-base-dataref-info-extract

*Ground truths.* **SciLite** [4]: automatically derived Europe PMC text-mining annotations, filtered to remove non-data subtypes (GO terms, RefSNP, etc.). **Gold GT**: manually curated, 388 paper–dataset links across all 249 test articles.

*Metrics.* Macro-averaged precision (P), recall (R), and F1 per article [1]. For analysis of model reliability, we additionally use the anchoring audit (§3.1) to characterize non-grounded outputs.

## 5 RESULTS

Table 2 reports precision, recall, and F1 for all model–configuration combinations on the 249-article held-out test set. Column $n$ is the number of articles for which at least one citation was extracted.

**Table 2: Results on the 249-article held-out test set. $n$ = articles with ≥1 citation extracted. Best per column in bold.**

| Model | Config | P | SciLite GT R | F1 | P | Gold GT R | F1 |
| --- | --- | --- | --- | --- | --- | --- | --- |
| Flan-t5-ft | base | .138 | .267 | .182 | .278 | .480 | .352 |
| | S3 | **.276** | .493 | **.354** | **.404** | .805 | **.538** |
| | RS3 | .308 | .583 | .404 | .426 | **.915** | .581 |
| Haiku-4.5 | base | .053 | .260 | .087 | .100 | .489 | .166 |
| | S3 | .107 | .483 | .176 | .151 | .790 | .253 |
| | RS3 | .116 | .558 | .193 | .155 | .882 | .264 |
| GPT-5-mini | base | .048 | .278 | .083 | .074 | .501 | .129 |
| | S3 | .088 | .501 | .150 | .106 | .803 | .188 |
| | RS3 | .103 | .588 | .175 | .116 | .906 | .206 |
| Gemini-3.5-flash | base | .071 | .285 | .114 | .117 | .517 | .191 |
| | S3 | .166 | .582 | .259 | .179 | .891 | .298 |
| | RS3 | .149 | **.602** | .239 | .163 | .914 | .277 |

### 5.1 RQ1: Can RTR be high-recall without full-document LLMs?

Across models, adding semantic retrieval (S3) yields the largest recall gains over the rule-based baseline, and adding regex anchoring (RS3) further improves recall. For Flan-T5, Gold-GT recall rises from 0.480 (base) to 0.805 (S3) and to 0.915 (RS3), while Gold F1 improves from 0.352 to 0.538 and to 0.581. On SciLite, Flan-T5 recall increases from 0.267 (base) to 0.493 (S3) and to 0.583 (RS3), with SciLite F1 increasing from 0.182 to 0.354 and to 0.404.

The same trend holds for commercial models. For Gemini-3.5-flash, Gold-GT recall rises from 0.517 (base) to 0.891 (S3) and to 0.914 (RS3), while SciLite recall rises from 0.285 (base) to 0.582 (S3) and to 0.602 (RS3). For GPT-5-mini, Gold-GT recall increases from 0.501 (base) to 0.803 (S3) and to 0.906 (RS3). These results confirm that recall losses previously observed for RTR are primarily driven by missed evidence in retrieval; semantic retrieval and regex anchoring substantially reduce this failure mode.

### 5.2 RQ2: Can a fine-tuned small model compete with commercial APIs?

At RS3, Flan-T5 achieves the best Gold-GT F1 (0.581) and precision (0.426). Commercial models attain comparable Gold-GT recall at

RS3 (Gemini-3.5-flash 0.914; GPT-5-mini 0.906; Haiku 0.882), but at substantially lower precision (0.163, 0.116, and 0.155 respectively), yielding lower Gold-GT F1 (0.277, 0.206, and 0.264). On SciLite, Gemini-3.5-flash attains the highest recall at RS3 (0.602), while Flan-T5 attains the best SciLite F1 at RS3 (0.404). Overall, the fine-tuned model provides a favorable quality–cost tradeoff: it runs locally at zero marginal API cost and, under the same retrieval configuration, yields substantially higher precision than commercial APIs.

## 6 DISCUSSION

*Generalization beyond GEO and PXD..* The strong Gold-GT recall (0.92 at RS3) on a test set spanning dozens of repository types confirms that the RTR pipeline generalizes well beyond the GEO/PXD-dominated evaluation of prior work, as long as target repositories are listed in the ontology.

*Cost and scalability.* The fine-tuned T5 runs on a single GPU at zero marginal cost per article, making it practical for large-scale sweeps over PubMed Central's open-access corpus. Commercial APIs introduce per-token costs and rate limits but can produce richer output (e.g., citation type). Our results suggest that without task-specific calibration, commercial models over-extract and dilute precision relative to the fine-tuned model.

*Limitations and next steps.* Regex patterns require maintenance as repository formats evolve. Coverage remains bounded by the ontology; SciLite can be used as a scalable signal to identify missing repository patterns and prioritize ontology enrichment. Future work includes automated ontology expansion via an iterative enrichment loop.

## AUTHORS

**Pietro Marini** (Data Tecnica International / NYU, *data management*) is a data engineer and researcher working on LLM-based information extraction from scientific literature. He developed the Data Gatherer system and leads its ongoing evaluation.

**Mike A. Nalls** (Data Tecnica International / NIH Center for Alzheimer's Research, *biomedical*) is Founder/Consultant at Data Tecnica International and Data Science Lead at NIH's Center for Alzheimer's Research, with expertise in large-scale genomic data analysis and biomedical data infrastructure.

**Mette Peters** (Division of Neuroscience, NIA/NIH, *biomedical*) is Senior Advisor to the Division of Neuroscience at the NIA/NIH, with over 25 years of experience in data science, data management architectures, data security, data sharing policies, and FAIR biomedical data across federal, industry, and nonprofit settings.

**Nicole Contaxis** (New York University, *biomedical*) is a data librarian and informaticist at NYU specializing in research data management, data sharing policy, and biomedical metadata standards.

**Juliana Freire** (New York University, *data management*) is Institute Professor at NYU Tandon and co-founder of the VIDA Center. She develops methods and systems for trustworthy data analysis, including the BDI-Kit data harmonization framework for biomedical data.

**Aécio Santos** (New York University, *data management*) is a research scientist at NYU VIDA working on data discovery, integration, and large-scale information extraction from scientific documents.

## ACKNOWLEDGMENTS

This work was supported in part by the NIH National Institute on Aging and NYU Tandon School of Engineering.

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
