# OpenReview forum: "Data Gatherer v0.2: High-Recall Retrieve-then-Read via Semantic Retrieval, Regex Anchoring, and Fine-Tuned T5"
_VLDB.org/2026/Workshop/BioDMS — BioDMS 2026 ProjectTalk_

### Official Review · Reviewer_AeJg · 2026-06-11

**Summary:**

The project describes an extension of an existing tool, DataGatherer, that was published in 2025 and retrieves dataset information from scientific articles. In particular, the retrieve-then-read approach was improved by including semantic ranking, regex anchoring and a fine-tuned small(er) language model instead of commercial LLMs for processing sections extracted based on rulesets. Its output can be used for better overviews on dataset usage, finding secondary datasets etc.

**Confidence Of Review:**

1

**Detailed Feedback Points:**

-	The benchmark is good and systematic, although the article reads a bit technical.
-	Providing open-source code and featuring smaller models instead of (computationally) expensive models is great and dearly needed.
-	Overall, the suggestions in the project are by nature (extension of parts of an existing tool) rather incremental than ground-breaking.

**Relevance For Biodms:**

2

---

### Official Review · Reviewer_n4tL · 2026-06-16

**Summary:**

In this submission, the authors present DataGatherer v0.2, an extension of their previous work on a system automatically extracting dataset references from PubMed Central papers. They include an improved method for extracting relevant article sections for further processing, which lowers the cost of full processing, and improves precision compared to the previous version.

**Confidence Of Review:**

2

**Detailed Feedback Points:**

Strong:
- comparison to commercial APIs is interesting and insightful
- useful nail and a hammer, which is an extension of previous work
- a clear, well-motivated improvement over the previously available approach

Neutral:
- maybe hard to imagine at a bona fide biomedical journal, but rather an interdisciplinary one (can imagine usefulness across multiple subfields of bioinformatics, medical informatics etc). I don't personally think this is a problem, although the "ideal" submission would have a slightly different profile according to the call for contributions
- thank you for openly sharing all code and artifacts, as per best practices of the field!

Opportunities for improvement:
- Could you please clarify if training code is also available in the git repo? A brief search seems to suggest so (which is great) but the manuscript only mentions evaluation code
- Table 2 - in some cases it's not the best value on bold, but e.g. second-best one. The lines grouping performance metrics accoring to a ground truth source seem to be misaligned
- an ablation study with just regex (R), without semantic retrieval (S3) would be needed to understand whether the performance improvement of RS3 over S3 comes from the combination of both, or just regex itself
- the "nail" could be motivated in more depth - why is large scale retrieval of dataset id from PMC papers useful? What could be a potential practical use case, what specific filters can be used to retrieve a subset of useful datasets? I assume extracting "all PMC datasets" is probably not exactly what the authors have in mind, but a clarification would help, especially taking an interdisciplinary audience into account
- it seems to be a relatively incremental extension of the previous version of DataGatherer. It is a meaningful improvement of a tool that is still under active development, but not fully sure if, by itself, this  submission would already be a good match for the most selective venues like top-tier data management conferences and a high-impact biomedical journals that the call for contributions expects. Perhaps it's a better match for the lightning talk?

**Relevance For Biodms:**

2

---

### Official Review · Reviewer_Min7 · 2026-06-20

**Summary:**

The paper introduces a thoughtful revision of the entire RTR pipeline rather than offering a single incremental improvement. The addition of semantic retrieval to overcome the limitations of fixed rules and the use of regex to catch identifiers in unusual sections represent a robust and systemic advancement. Furthermore, the decision to openly release models, data, and code aligns with the goals of reproducibility and accessibility.

**Confidence Of Review:**

3

**Detailed Feedback Points:**

> The system is heavily dependent on the repository ontology and regex patterns. This can create a maintenance bottleneck and limit scalability to new repositories or evolving identifier patterns.

> The discussion on how to sustainably maintain and update this ontology remains incomplete. In particular, the suggestion to use SciLite is valid, but the details of the "iterative enrichment loop" are not clearly discussed.

> It is necessary to motivate the choice of k=3 through experimentation or an heuristic.

> Could authors discuss the impact of this parameter on the trade-off between recall and precision?

> Regarding the outputs, have the authors considered using a post-hoc verification mechanism, such as a second model or a simple regex filter, to discard or revise these predictions?

> In my opinion, the title is a bit vague. I suggest focusing more on the technical contribution.

**Relevance For Biodms:**

3